# FL Games: A Federated Learning Framework for Distribution Shifts

**Sharut Gupta**
Mila, Universite de Montreal
Imagia Cybernetics Inc.
Indian Institute of Technology Delhi
`sharut.gupta@mila.quebec`

**Kartik Ahuja**
Mila, Universite de Montreal
`kartik.ahuja@mila.quebec`

**Mohammad Havaei**
Imagia Cybernetics Inc.
`mohammad@imagia.com`

**Niladri Chatterjee**
Indian Institute of Technology Delhi
`niladri@maths.iitd.ac.in`

**Yoshua Bengio**
Mila, Universite de Montreal
`yoshua.bengio@mila.quebec`

## Abstract

Federated learning aims to train predictive models for data that is distributed across clients, under the orchestration of a server. However, participating clients typically each hold data from a different distribution, which can yield to catastrophic generalization on data from a different client, which represents a new domain. In this work, we argue that in order to generalize better across non-i.i.d. clients, it is imperative to only learn correlations that are stable and invariant across domains. We propose FL GAMES, a game-theoretic framework for federated learning that learns causal features that are invariant across clients. While training to achieve the Nash equilibrium, the traditional best response strategy suffers from high-frequency oscillations. We demonstrate that FL GAMES effectively resolves this challenge and exhibits smooth performance curves. Further, FL GAMES scales well in the number of clients, requires significantly fewer communication rounds, and is agnostic to device heterogeneity. Through empirical evaluation, we demonstrate that FL GAMES achieves high out-of-distribution performance on various benchmarks.

## 1 Introduction

With the rapid advance in technology and growing prevalence of smart devices, Federated Learning (FL) has emerged as an attractive distributed learning paradigm for machine learning models over networks of computers Kairouz et al. [2019], Li et al. [2020], Bonawitz et al. [2019]. In FL, multiple sites with local data, often known as *clients*, collaborate to jointly train a shared model under the orchestration of a central hub called the *server* while keeping their data private.

While FL serves as an attractive alternative to centralized training because the client data does not need to move to the server, *statistical heterogeneity* is a key challenge in its optimization. While one of the most popular algorithms in this setup, Federated Averaging (FEDAVG) McMahan et al. [2017] delivers huge communication gains in i.i.d. (independent and identically distributed) setting, its

Workshop on Federated Learning: Recent Advances and New Challenges, in Conjunction with NeurIPS 2022 (FL-NeurIPS'22). This workshop does not have official proceedings and this paper is non-archival.

performance on non-i.i.d. clients is an active area of research. As shown by Karimireddy et al. [2020], client heterogeneity has direct implications on the convergence of FEDAVG since it introduces a *drift* in the updates of each client with respect to the server model. While recent works Li et al. [2019], Karimireddy et al. [2020], Yu et al. [2019], Wang et al. [2020], Li et al. [2020], Lin et al. [2020], Li and Wang [2019], Zhu et al. [2021] have tried to address client heterogeneity through constrained gradient optimization and knowledge distillation, most did not tackle the underlying distribution shift. These methods can at best generalize to interpolated domains and fail to extrapolate well, i.e., generalize to new extrapolated domains [1].

Over the past year, there has been a surge in interest in bringing the machinery of causality into machine learning Arjovsky et al. [2019], Ahuja et al. [2020], Schölkopf [2019], Ahuja et al. [2021], Parascandolo et al. [2020], Robey et al. [2021], Krueger et al. [2021], Rahimian and Mehrotra [2019]. However, despite their success, they suffer from key limitations which are unsuitable for deployment in a real-world setup.

Since FL typically consists of a large number of clients, it is natural for data at each client to represent different annotation tools, measuring circumstances, experimental environments, and external interventions. Inspired by this idea and by the recent progress in causal machine learning, we draw connections between OOD generalization and robustness across heterogeneous clients in FL. To date, only two scientific works Francis et al. [2021], Tenison et al. [2021] have incorporated the learning of invariant predictors in order to achieve strong generalization in FL. The former adapts masked gradients Parascandolo et al. [2020] and the latter builds on IRM to exploit invariance and improve leakage protection in FL. While IRM lacks theoretical convergence guarantees, failure modes of Parascandolo et al. [2020] like formation of dead zones and high sensitivity to small perturbations Shahtalebi et al. [2021] are also issues when it is applied in FL, rendering it unreliable.

In this study, we consider IRM GAMES Ahuja et al. [2020] from the OOD generalization literature since its formulation shows resemblance to the standard FL setup. However, as discussed above, IRM GAMES too encounters a few fundamental challenges not just specific to FL but also in a generic ML framework. We take a step towards fixing these limitations and addressing the challenge of client heterogeneity under distribution shifts in FL from a causal viewpoint. Specifically, we propose Federated Learning Games (FL GAMES) for learning causal representations which are stable across clients. We summarize the our main contributions below.

- We propose a new framework called FL GAMES for learning causal representations that are invariant across clients in a federated learning setup.
- The underlying sequential game theoretic framework in IRM GAMES causes the time complexity of FL algorithm to scale linearly with the number of clients. Inspired from the game theory literature, we equip our algorithm to allow parallel updates across clients, further resulting in superior scalability.
- IRM GAMES exhibits large oscillations in the performance metrics as the training progresses, making it difficult to define a valid stopping criterion. Using ensembles over client's historical actions, we demonstrate that FL GAMES appreciably smoothens these oscillations.
- The convergence rate of IRM GAMES is slow and hence directly impacts systems with speed or communication cost as a primary bottleneck. By increasing the local computation at each client, we show that FL GAMES exhibits high communication efficiency
- Empirically, we show that the performance of the invariant predictors found by our approach on unseen OOD clients improves significantly over state-of-art prior works.

## 2 Background

### 2.1 Causality in Machine Learning

Consider a multi-source domain generalization task, where the goal is to learn a robust set of parameters that generalize well to unseen (test) environments $\mathcal{E}_{all} \supset \mathcal{E}_{tr}$, given a set of $m$ training domains (or environments) $\mathcal{E}_{tr}$. A popular algorithm, Invariant Risk Minimization (IRM) Arjovsky et al. [2019] does so by defining the Invariance principle

---

[1]Similar to Krueger et al. [2021], we define interpolated domains as the domains which fall within the convex hull of training domains and extrapolated domains as those that fall outside of that convex hull.

**Definition 1** *A data representation $\phi : \mathcal{X} \to \mathcal{Z}$ elicits an invariant predictor $w \circ \phi$ across environments $\mathcal{E}_{tr}$ if there is a classifier $w : \mathcal{Z} \to \mathcal{Y}$ simultaneously optimal for all environments i.e. $w \in \arg\min_{w' \in \mathcal{Z} \to \mathcal{Y}} R^e(w' \circ \phi), \forall e \in \mathcal{E}_{tr}$*

Invariant Risk Minimization Games (IRM GAMES) propose a game theoretic formulation for finding $(\phi, w)$ that satisfy the invariance principle. It endows each environment with its own predictor $w^k \in \mathcal{H}_w$ and aims to train an ensemble model $w^{av}(z) = \frac{1}{|\mathcal{E}_{tr}|} \sum_{k=1}^{|\mathcal{E}_{tr}|} w^k(z)$ for each $z \in \mathcal{Z}$ s.t. $w^{av}$ satisfies the following optimization problem

$$\min_{w^{av}, \phi \in \mathcal{H}_\phi} \sum_k R^k(w^{av} \circ \phi) \text{s.t. } w^k \in \arg\min_{w'_k \in \mathcal{H}_w} R^k \left( \frac{1}{|\mathcal{E}_{tr}|} (w'_k + \sum_{\substack{q \in \mathcal{E}_{tr} \\ q \neq k}} w^q) \circ \phi \right), \forall k \in \mathcal{E}_{tr} \quad (1)$$

where $\mathcal{H}_\phi$, $\mathcal{H}_w$ are the hypothesis sets for feature extractors and predictors, respectively. The constraint in equation 1 is equivalent to the Nash equilibrium of a game with each environment $k$ as a player with action $w^k$, playing to maximize its utility $-R^k(w^{av}, \phi)$. The resultant game is solved using the best response dynamics (BRD) with clockwise updates (for more details, refer to the supplement) and is referred to as V-IRM GAMES. Fixing $\phi$ to an identity map in V-IRM GAMES is also shown to be very effective and is called F-IRM GAMES.

# 3 Federated Learning Games (FL GAMES)

OOD generalization is often typified using the notion of data-generating environments. Arjovsky et al. [2019] formalizes an environment as a data generating distribution representing a particular location, time, context, circumstances and so forth. This concept of data-generating environments can be related to FL by considering each client as producing data generated from a different environment. However despite this equivalence, existing OOD generalization techniques can't be directly applied to FL. Apart from the FL-specific challenges, these approaches also suffer from several key limitations in non-FL domains Rosenfeld et al. [2020], Nagarajan et al. [2020], further rendering them unfit for practical deployment. Thus developing causal inference models for FL which are inspired from invariant prediction in OOD generalization, are bound to inherit the failures of the latter.

In this work, we consider one such popular OOD generalization technique, IRM GAMES as it formulation bears close resemblance to a standard FL setup. However, as discussed, IRM GAMES too suffers from various challenges, which impede its deployment in a generic ML framework, specifically in FL. In the following section, we elaborate on each of these limitations and discuss the corresponding modifications required to overcome them. Further, inspired from the game theoretic formulation of IRM GAMES, we propose FL GAMES which forfeits it's failures and can recover causal mechanisms of the targets, while also providing robustness to changes in the distribution.

## 3.1 Challenges in Federated Learning

**Data Privacy.** Consider a FL system with $m$ client devices, $\mathcal{S} = \{1, 2, ..., m\}$. Let $N_k$ denote the number of data samples at client device $k$, and $\mathcal{D}_k = \{(x_i^k, y_i^k)\}_{i=1}^{N_k}$ as it's labelled dataset. The constraint of each environment in IRM GAMES can be used to formulate the local objective of each client. In particular, each client $k \in \mathcal{S}$ now serves as a player, competing to learn $w^k \in \mathcal{H}_w$ by optimizing its local objective, i.e. $w^k \in \arg\min_{w'_k \in \mathcal{H}_w} R^k \left( \frac{1}{|\mathcal{S}|} (w'_k + \sum_{\substack{q \in \mathcal{E}_{tr} \\ q \neq k}} w^q) \circ \phi \right), \forall w'_k \in \mathcal{H}_w$ However, the upper-level objective of IRM GAMES requires optimization over the dataset pooled together from all environments. Centrally hosting the data at the server or sharing it across clients contradicts the objective of FL. Hence, we propose using FEDSGD McMahan et al. [2017] to optimize $\phi$. Specifically, each client $k$ computes and broadcasts $g^k = \nabla R^k \left( \frac{1}{|\mathcal{S}|} (w'_k + \sum_{q \neq k} w^k) \circ \phi \right)$ to the server, which then aggregates these gradients and applies the update rule $\phi_{t+1} = \phi_t - \eta \sum_{k \in \mathcal{S}} \frac{N_k}{N} g^k$ where $g^k$ is computed over $C\%$ batches locally and $N = \sum_{k \in S} N_k$. We call this approach V-FL GAMES. The variant with $\phi = I$ is called F-FL GAMES.

**Sequential dependency** As discussed, IRM GAMES poses IRM objective as finding the Nash equilibrium of an ensemble game across environments and adopts the classic best response dynamics (BRD) algorithm to compute it. This approach is based on playing clockwise sequences wherein players take turns in a fixed cyclic order, with only one player being allowed to change its action at any given time $t$ (details in the supplement). This linear scaling of time complexity with the number of players poses a major challenge solving the game in FL. We modify the classic BRD algorithm by allowing simultaneous updates at any given round $t$. However, a client now best responds to the optimal actions played by it's opponent clients in round $t - 1$ instead of $t$. We refer to this approximation of BRD in F-FL GAMES and V-FL GAMES as *parallelized* F-FL GAMES and, *parallelized* V-FL GAMES respectively.

**Oscillations.** As demonstrated in Ahuja et al. [2020], when a neural network is trained using the IRM GAMES objective (equation 1), the training accuracy initially stabilizes at a high value and eventually starts to oscillate. The explanation for these oscillations attributes to the significant difference among the data of the training environments. With model's performance metrics oscillating to and from at each step, defining a reasonable stopping criterion becomes challenging. As shown in various game theoretic literature Herings and Predtetchinski [2017], Barron et al. [2010], Fudenberg et al. [1998], Ge et al. [2018], BRD can often oscillate. Computing the Nash equilibrium for general games is non-trivial and is only possible for a certain class of games (e.g., concave games) Zhou et al. [2017]. Thus, rather than alleviating oscillations completely, we propose solutions to reduce them significantly to better target valid stopping points. We propose a two-way ensemble approach wherein apart from maintaining an ensemble across clients ($w^{av}$), each client $k$ also responds to the ensemble of historical models (memory) of its opponents. Formally, we maintain queues (a.k.a. buffer) at each client which store its historically played actions. In each iteration, a client best responds to a uniform distribution over the past strategies of its opponents. The global objective at the server remains unchanged. Mathematically, the new local objective of each client $k \in \mathcal{E}_{tr}$ can be stated as

$$w^k \in \operatorname*{arg\,min}_{w'_k \in \mathcal{H}_w} R^k \left( \frac{1}{|\mathcal{S}|} (w'_k + \sum_{\substack{q \in \mathcal{S} \\ q \neq k}} w^q + \sum_{\substack{p \in \mathcal{S} \\ p \neq k}} \frac{1}{|\mathcal{B}_p|} \sum_{j=1}^{|\mathcal{B}_p|} w_j^p) \circ \phi \right) \tag{2}$$

where $\mathcal{B}_q$ denotes the buffer at client $q$ and $w_j^q$ denotes the $j$th historical model of client $q$. As the buffer reaches its capacity, it is renewed based on first in first out (FIFO) manner. Note that this approach doesn't result in any communication overhead since a running sum over historical strategies can be calculated in $\mathcal{O}(1)$ time by maintaining a prefix sum. This variant is called F-FL GAMES (SMOOTH) or V-FL GAMES (SMOOTH) based on the constraint on $\phi$.

**Convergence speed.** As discussed, FL GAMES has two variants: F-FL GAMES and, V-FL GAMES with the former being an approximation of the latter ($\phi = I$). While both the approaches exhibit superior performance on a variety of benchmarks, the latter has shown its success in a variety of large scale tasks like language modeling. Peyrard et al. [2021]. Despite being theoretically grounded, V-IRM GAMES suffers from slower convergence due to an additional round for optimization of $\phi$.

Typically, clients (e.g. mobile devices) posses fast processors and computational resources and have datasets that are much smaller compared to the total dataset size. Hence, utilizing additional local computation is essentially free compared to communicating with the server. To improve the efficiency of our algorithm, we propose replacing the stochastic gradient descent (SGD) over $\phi$ by a full-batch gradient descent (GD). Intuitively, now at each gradient step, the resultant $\phi$ takes large steps in the direction of its global optimum, resulting in fast training. Note that the classifiers at each client are still updated over one mini-batch. This variant of FL GAMES is referred to V-FL GAMES (FAST).

## 4 Experiments and Results

In Ahuja et al. [2020], IRM GAMES was tested over a variety of benchmarks including COLORED MNIST, COLORED FASHION MNIST and COLORED DSPRITES dataset. We utilize the same datasets for our experiments. Additionally, we create another benchmark, SPURIOUS CIFAR10. Details on each of the datasets can be found in the supplement. We report the mean performance of various baselines over 5 runs.

We compare these algorithmic variants across fixed and variable $\phi$ separately as shown in the Table 1. Clearly, across all benchmarks, the FL baselines FedSGD, FedAVG McMahan et al. [2017], FedBN Li et al. [2021] and FedPROX Li et al. [2020] are unable to generalize to the test set, with FedBN exhibiting superior performance compared to the others. Intuitively, these approaches latch onto the spurious features to make prediction, hence leading to poor generalization over novel clients. From Table 1, we observe that all modifications in FL Games individually achieve high testing accuracy, hence eliminating the spurious correlations unlike state-of-the-art FL techniques. Further, on Colored Dsprites, we discover that the mean performance of V-FL Games is superior to all algorithms with fixed representation ($\phi = I$). This accentuates the importance of V-FL Games over F-FL Games, especially over complex and larger datasets where learning $\phi$ becomes imperative.

Table 1: Comparison of methods in terms of training and testing accuracy (mean ± std deviation).'Seq.' and 'Par.' are abbreviations for sequential and parallel respectively.

| | | Algorithm | Colored MNIST | | Colored Fashion MNIST | | Spurious CIFAR10 | | Colored Dsprites | |
| --- | --- | --- | --- | --- | --- | --- | --- | --- | --- | --- |
| | | | Train Accuracy | Test Accuracy | Train Accuracy | Test Accuracy | Train Accuracy | Test Accuracy | Train Accuracy | Test Accuracy |
| Baselines | | FedSGD | 84.88 ± 0.16 | 10.45 ± 0.60 | 83.49 ± 1.22 | 20.13 ± 8.06 | 84.79 ± 0.17 | 12.57 ± 0.55 | 99.15 ± 1.10 | 24.12 ± 2.00 |
| | | FedAVG | 84.45 ± 2.69 | 12.52 ± 4.34 | 86.23 ± 0.63 | 13.33 ± 2.07 | 85.41 ± 1.45 | 13.11 ± 1.82 | 99.21 ± 1.35 | 22.56 ± 2.34 |
| | | FedBN | 99.75 ± 0.11 | 47.16 ± 3.76 | 99.79 ± 0.17 | 41.24 ± 1.87 | 95.24 ± 2.34 | 25.16 ± 4.06 | 98.09 ± 1.45 | 25.19 ± 1.78 |
| | | FedPROX | 99.56 ± 0.38 | 29.31 ± 0.89 | 99.87 ± 0.12 | 29.41 ± 0.36 | 99.67 ± 0.13 | 24.76 ± 2.67 | 85.17 ± 1.95 | 11.12 ± 0.99 |
| Fixed | Seq. | F-FL Games | 55.76 ± 2.03 | 66.56 ± 1.58 | 75.13 ± 1.38 | 68.40 ± 1.83 | 50.36 ± 2.78 | 45.36 ± 4.33 | 53.98 ± 3.67 | 52.89 ± 4.41 |
| | | F-FL Games (Smooth) | 62.83 ± 5.06 | 66.83 ± 1.83 | 75.18 ± 0.37 | **71.81 ± 1.60** | 64.02 ± 2.08 | 45.54 ± 1.04 | 52.87 ± 3.30 | 61.45 ± 7.11 |
| | Par. | F-FL Games | 58.03 ± 6.22 | 67.14 ± 2.95 | 71.71 ± 8.23 | 69.73 ± 2.12 | 55.06 ± 2.04 | 52.07 ± 1.60 | 52.88 ± 2.78 | 56.50 ± 6.23 |
| | | F-FL Games (Smooth) | 61.07 ± 1.71 | **67.21 ± 2.98** | 72.81 ± 4.51 | 71.36 ± 4.19 | 56.98 ± 4.09 | **54.71 ± 2.13** | 53.65 ± 2.11 | **62.76 ± 5.97** |
| Variable | Seq. | V-FL Games | 56.40 ± 0.03 | 63.78 ± 1.58 | 69.90 ± 4.56 | **69.90 ± 1.31** | 61.72 ± 7.39 | 46.07 ± 6.01 | 51.36 ± 5.32 | 62.84 ± 7.20 |
| | | V-FL Games (Smooth+Fast) | 61.03 ± 3.11 | 65.81 ± 3.28 | 75.10 ± 0.48 | 69.85 ± 1.22 | 50.37 ± 4.97 | **50.94 ± 3.28** | 51.55 ± 3.20 | 68.23 ± 4.56 |
| | Par. | V-FL Games | 52.89 ± 8.03 | **68.34 ± 5.24** | 66.33 ± 9.39 | 69.85 ± 3.42 | 50.41 ± 3.31 | 50.43 ± 3.04 | 53.56 ± 4.91 | 65.87 ± 6.84 |
| | | V-FL Games (Smooth+Fast) | 63.11 ± 3.02 | 65.73 ± 1.53 | 71.89 ± 5.58 | 69.41 ± 5.49 | 45.83 ± 2.44 | 49.89 ± 5.66 | 54.25 ± 2.05 | **68.91 ± 6.47** |
| | | Optimal | 75 | 75 | 75 | 75 | 75 | 75 | 75 | 75 |

In all the above experiments, both of our end approaches: *parallelized* V-FL Games (Smooth +Fast) and *parallelized* F-FL Games (Smooth) are able to perform better than or at par with the other variants. *These algorithms were primarily designed to overcome the challenges faced by causal FL systems while retaining their original ability to learn causal features.* Hence, the benefits provided by these approaches in terms of 1) robust predictions; 2) scalability; 3) fewer oscillations and 4) fast convergence are not at the cost of performance. While the former is demonstrated by Table 1, the latter three are detailed in Section 4.2.

## 4.1 Interpretation of Learned Features

In order to explain the predictions of our parametric model, we use LIME Ribeiro et al. [2016] to learn an interpretable model around each prediction. Figure 1(a) shows LIME masks for both FedAVG (ii) and our method (iii). Clearly, the former focuses solely on the background to make the model prediction. However, the latter uses causal features in the image (like shape, stroke, edges and curves) along with some pixels from the background (noise) to make predictions. This demonstrates the reasoning behind robust generalization of FL Games as opposed to state-of-the-art FL techniques.

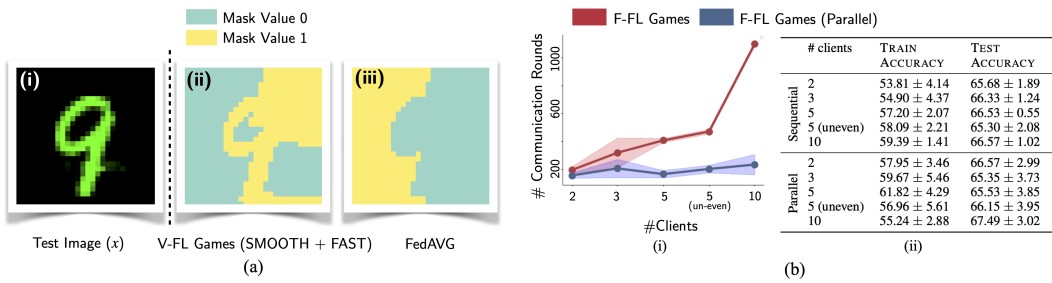

Figure 1: (a)(i) Test image used as input to the interpretable model; LIME mask corresponding to model prediction using (ii)V-FL Games (Smooth+Fast); (iii)FedAVG; (b) Comparison of F-FL Games and F-FL Games (Parallel) with increasing client in terms of (i) communication rounds; (ii) performance.

## 4.2 Ablation Analysis

We analyze the effect of each of our algorithmic modifications on the COLORED MNIST dataset. The results on other datasets are similar and are provided in the supplement.

### 4.2.1 Effect of Simultaneous BRD

We examine the effect of replacing the classic best response dynamics as in Ahuja et al. [2020] with the simultaneous best response dynamics. For the same, we use a more practical environment: (a) more clients are involved, and (b) each client has fewer data. Similar to Choe et al. [2020], we extended the COLORED MNIST dataset by varying the number of clients between 2 and 10. For each setup, we vary the degree of spurious correlation between 70% and 90%) for training clients and merely 10% in the testing set. A more detailed discussion of the dataset is provided in the supplement. For F-FL GAMES, it can be observed from Figure 1(b.i), as the number of clients in the FL system increase, there is a sharp increase in the number of communication rounds required to reach equilibrium. However, the same doesn't hold true for *parallelized* F-FL GAMES. Further, *parallelized* F-FL GAMES is able to reach a comparable or higher test accuracy as compared to F-FL GAMES with significantly lower communication rounds (refer to Figure 1(b.ii)).

### 4.2.2 Effect of a memory ensemble

As shown in Figure 2(b), compared to F-FL GAMES, F-FL GAMES (SMOOTH) reduces the oscillations significantly. While in the former, performance metrics oscillate at each step, the oscillations in the later are observed after an interval of roughly 50 rounds. Further, F-FL GAMES (SMOOTH) seems to envelop the performance curves of F-FL GAMES. As a result, apart from reducing the frequency of oscillations, F-FL GAMES (SMOOTH) also achieves higher testing accuracy compared to F-FL GAMES. The observations are consistent across the *parallelized* variants. Further, as observed from Figure 2(a), both sequential and parallel F-FL GAMES (SMOOTH) significantly reduce the number of rounds required to reach equilibrium, further underscoring the efficacy of our proposed methodology.

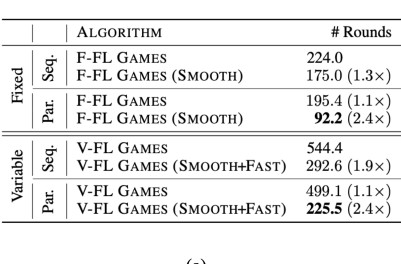

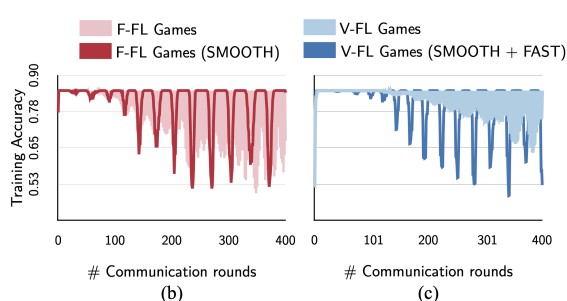

Figure 2: COLORED MNIST: (a) Comparison of F-FL GAMES and F-FL GAMES (Parallel) with increasing clients; Training accuracy of (b) F-FL GAMES and F-FL GAMES (SMOOTH) for a buffer size of 5; (c) V-FL GAMES and V-FL GAMES (SMOOTH+FAST) with buffer size as 5 versus the number of rounds;

### 4.2.3 Effect of using Gradient Descent (GD) for $\phi$

Communication costs are the principal constraints in FL setup. Edge devices like mobile phones and sensor are bandwidth constrained and require more power for transmission and reception as compared to remote computation. As observed from Figure 2(c), V-FL GAMES (SMOOTH+FAST) is able to achieve significantly higher testing accuracy in fewer rounds as compared to V-FL GAMES. Consistent results are also reported in Figure 2(a), where both sequential and parallel variants of V-FL GAMES (SMOOTH+FAST) result in a significant improvement ($\sim$2×) in the number of rounds.

## 5 Conclusion

In this work, we develop a novel framework based on Best Response Dynamics (BRD) training paradigm to learn invariant predictors across clients in Federated learning (FL). Inspired from Ahuja et al. [2020], the proposed method called Federated Learning Games (FL GAMES) learns causal

representations which have good out-of-distribution generalization on new train clients or test clients unseen during training. We investigate the high frequency oscillations observed using BRD and equip our algorithm with a memory of historical actions. This results in smoother performance metrics with significantly lower oscillations. FL GAMES exhibits high communication efficiency as it allows parallel computation, scales well in the number of clients and results in faster convergence. Given the impact of FL in medical imaging, we plan to test our framework over medical benchmarks. Future directions include theoretically analyzing the smoothed best response dynamics, as it might have potential implications for other game-theoretic based machine learning frameworks.

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

## A   Appendix

## B   Game Theory Concepts

We define some basic game theory notations that will be used later. Let $\Gamma = (N, \{S_k\}_{k \in N}, \{u_k\}_{k \in N})$ be the tuple representing a normal form game, where $N$ denotes the finite set of players. For each player $k$, $S_k = \{s_0^k, s_1^k, ... s_m^k\}$ denotes the pure strategy space with $m$ strategies and $u_k(s_k, s_{-k})$ denotes the payoff function of player $k$ corresponding to strategy $s_k$. Here, an environment for player $k$ is $s_{-k}$, a set containing strategies taken by all players but $k$ and $S_{-k}$ denotes the space of strategies of the opponent players to $k$. $S = \prod_{i \in N} S_i$ denotes the joint strategy set of all players. A game $\Gamma$ is said to be finite if $S$ is finite and is continuous if $S$ is uncountably infinite.

While a pure strategy defines a specific action to be followed at any time instance, a mixed strategy of player $k$, $\sigma_k = \{p_k(s_0^k), p_k(s_1^k), ... p_k(s_m^k)\}$ is a probability distribution over a set of pure strategies, where $\sum_{j=1}^m p_k(s_j^k) = 1$. The expected utility of a mixed strategy $u_k(\sigma_k, \sigma_{-k})$ for player $k$ is the expected value of the corresponding pure strategy payoff i.e.

$$\mathbb{E}(u_k(\sigma_k, \sigma_{-k})) = \sum_{s_k \in S_k} \sum_{s_{-k} \in S_{-k}} u_k(s_k, s_{-k}) p_k(s_k) p_{-k}(s_{-k}), \forall \sigma_k \in \tilde{S}_k \tag{3}$$

where $\tilde{S}_k$ corresponds to the mixed strategy space of player $k$.

**Best response (BR).** A mixed strategy $\sigma_k^*$ for player $k$ is said to be a best response to it's opponent strategies $\sigma_{-k}$ if

$$\mathbb{E}(u_k(\sigma_k^*, \sigma_{-k})) \geq \mathbb{E}(u_k(\sigma_k, \sigma_{-k})), \forall \sigma_k \in \tilde{S}_k.$$

**Nash equilibrium.** A mixed strategy profile $\sigma^* = \{\sigma_1^*, \sigma_2^*, ... \sigma_N^*\}$ is a Nash equilibrium if for all players $k$, $\sigma_k^*$ is the best response to the strategies played by it's opponent players i.e $\sigma_{-k}^*$.

**Best response dynamics (BRD).** BRD is an iterative algorithm in which at each time step, a player myopically plays strategies that are best responses to the most recent known strategies played by it's opponents previously. Based on the playing sequence across layers, BRD can be classified into three broad categories: BRD with clockwise sequences, BRD with simultaneous updating and BRD with random sequences. For this study, we focus only on the first two playing schedules. Let the function seq : $\mathbb{N} \to \mathcal{P}[N]$ denote a playing sequence which determines the set of players whose turn it is to play at each time period $t \in \mathbb{N}$. Here, $\mathcal{P}[N]$ denotes the power set of $\{1, 2, ... N\}$ players and $\mathbb{N}$ be the set of natural numbers $\{1, 2, ...\}$. By BRD, at each time step $t$, $\forall i \in \text{seq}(t)$, action taken by player $i$ i.e. $a_i^t$ is the best response to it's current environment i.e $a_{-i}^t$.

- BRD with clockwise sequences: In this playing sequence, players take turns according to a fixed cyclic order and only one player is allowed to change it's action at any given time $t$. Specifically, the playing sequence is defined by $\text{seq}(t) = 1 + (t - 1) \mod n$. Since only a single player is allowed to play at any given time $t$, $a_{\text{-seq}(t)}^t = a_{\text{-seq}(t)}^{t-1}$.

- BRD with simultaneous updating: In this playing sequence, $\text{seq}(t)$ chooses a non empty subset of players to participate in round $t$. However, for each player $i \in \text{seq}(t)$, the optimal action chosen $a_i^t$ depends on the knowledge of the latest strategy of it's opponents.

## C   Datasets

The MNIST dataset consists of handwritten digits, with a total of 60,000 images in the training set and 10,000 images in the test set Deng [2012]. These images are black and white in colour and form a subset of the larger collection of digits called NIST. Each digit in the dataset is normalized in size to centre fit in the fixed size image of size 28×28. It is then anti-aliased to introduce appropriate gray-scale levels.

### C.1   COLORED MNIST

We modify the MNIST dataset in the exact same manner as in Arjovsky et al. [2019]. Specifically, Arjovsky et al. [2019] creates the dataset in a way that it contains both the invariant and spurious

features according to different causal graphs. Spurious features are introduced using colors. Digits less than 5 (excluding 5) are attributed with label 0 and the rest with label 1. The dataset is divided across three clients, out of which two serve as training and one as testing. The 60,000 images from MNIST train set are divided equally amongst the two training clients i.e. each consists of 30,000 samples. The testing set contains the 10,000 images from the MNIST test set. Preliminary noise is added to the label to reduce the invariant correlation. Specifically, the initial label ($\tilde{y}$) of each image is flipped with a probability $\delta_k$ to construct the final label $y$. The final label $y$ of each image is further flipped with a probability $p_k$ to construct it's color code ($z$). In particular, the image is colored red, if $z = 1$ and green if $z = 0$. The flipping probability which defines the color coding of an image, $p_k$ is 0.2 for client 1, 0.1 for client 2 and 0.9 for the test client. The probability $\delta_k$ is fixed to 0.25 for all clients $k$. The above choice is defined in a way that the mean degree of label-color (spurious) correlation ($1 - p_k, \forall k$) is more than the average degree of invariant correlation ($1 - \delta_k, \forall k$). A sample batch of images elucidating the above construction is shown in Figure 3.

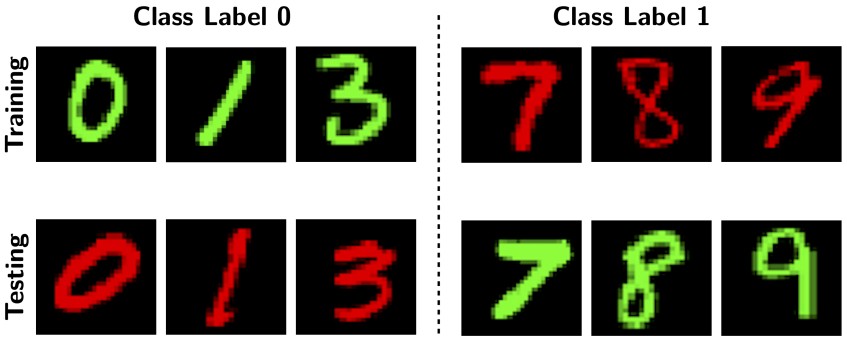

Figure 3: COLORED MNIST: Illustration of samples containing high spurious correlation between label and color during training. While testing, this correlation is significantly reduced as label 0 is highly correlated with the color mapped to label 1 and vice-versa.

## C.2 COLORED FASHION MNIST

We use the exact same environment for creating COLORED FASHION MNIST as in Ahuja et al. [2020]. The data generating process of COLORED FASHION MNIST is motivated from that of COLORED MNIST in a way that it possesses spurious correlations between the label and the colour. Fashion MNIST consists of images from a variety of sub-categories under the two broad umbrellas of clothing and footwear. Clothing items include categories like: "t-shirt", "trouser", "pullover", "dress", "shirt" and "coat" while the footwear category includes "sandal", "sneaker", "bag" and "ankle boots". Similar to COLORED MNIST, the train dataset is equally split across two clients (30,000 images each) and the entire test set is attributed to the test client. Preliminary labels for binary classification are constructed such $\tilde{y} = 0$ for "t-shirt", "trouser", "pullover", "dress", "coat", "shirt" and $\tilde{y} = 1$: "sandle", "sneaker" and "ankle boots". Next, we add noise to the preliminary label by flipping $\tilde{y}$ with a probability $\delta_k = 0.25, \forall k$ to construct the final label $y$. We next flip the final label with a probability $p_k$ to designate a color ($z$), with $p_1 = 0.2$ for the first client, $p_2 = 0.1$ for the second client and $p_3 = 0.9$ for the test client. The image is colored red, if $z = 1$ and green if $z = 0$. A sample batch of images elucidating the above construction is shown in Figure 4.

## C.3 SPURIOUS CIFAR10

In this setup, we modify the CIFAR-10 dataset similar to the COLORED MNIST dataset. Instead of coloring the images, we use a different mechanism based on the spatial location of a synthetic feature to generate spurious features. CIFAR10 dataset consists of 60,000 images from 10 classes including "airplane", "automobile", "bird", "cat", "deer", "dog", "frog", "house", "ship", "truck". The original dataset is relabelled to create a binary classification task between motor and non-motor objects. All images corresponding to the label "frog" are discarded to ensure a similar samples count for the two classes. Similar to COLORED MNIST, the train dataset is equally split across two clients and the entire test set is attributed to the test client. Preliminary labels for binary classification are constructed such $\tilde{y} = 0$ for "airplane", "automobile", "ship", "truck" and $\tilde{y} = 1$: "bird", "cat", "deer","dog" and

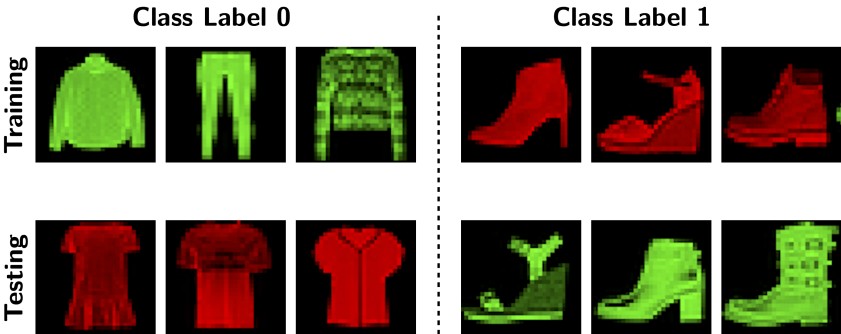

Figure 4: COLORED MNIST: Illustration of samples containing high spurious correlation between label and color during training. While testing, this correlation is significantly reduced as label 0 is highly correlated with the color mapped to label 1 and vice-versa.

"horse". Next, we add noise to the preliminary label by flipping $\tilde{y}$ with a probability $\delta_k = 0.25, \forall k$ to construct the final label $y$. We next flip the final label with a probability $p_k$ to designate a positional index ($z$), with $p_1 = 0.2$ for the first client, $p_2 = 0.1$ for the second client and $p_3 = 0.9$ for the test client. This index defines the spatial location of a 5×5 black patch in the image. An index value of $0$ ($z = 0$) specifies the patch at the top left corner of the image, while an index value of 1 ($z = 1$) corresponds to a black patch over the top-right corner of the image. Based on the choice of flipping probabilities, images in the training set are spuriously correlated to the position of the patch in the image. Such correlation does not exist while testing. A sample batch of images elucidating the above construction is shown in Figure 5.

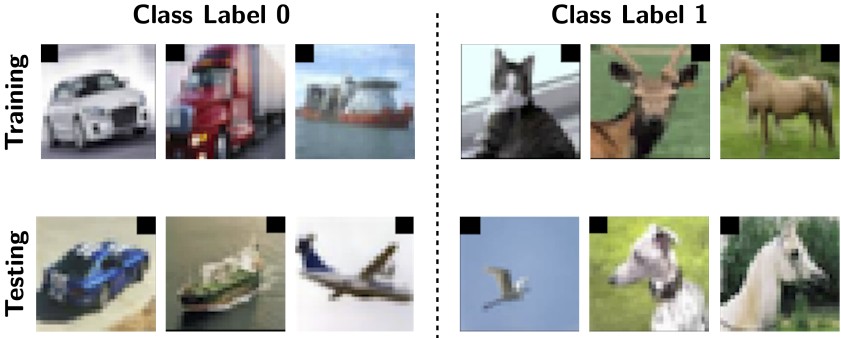

Figure 5: SPURIOUS CIFAR10: Illustration of samples containing high spurious correlation between labels and spatial location of the 5×5 black patch during training. While testing, this correlation is significantly reduced as label 0 is now highly correlated with the spatial location corresponding to label 1 and vice-versa.

### C.4 COLORED DSPRITES

To construct this FL setup, we modify the Desprites dataset[2](737,280 images) as inAhuja et al. [2020]. The data generating process of COLORED DSPRITES is motivated from that of COLORED MNIST in a way that it possesses spurious correlations between the label and the colour. Desprites dataset consists of 2D shapes procedurally generated from 6 ground truth independent latent factors. These factors are color, shape, scale, rotation, x and y positions of a sprite. The shape further belongs to a set of three different categories: square, ellipse or heart. We convert this multi-label and multi-class task into binary classification problem. Specifically, we consider classification between a circle and a square. Similar to COLORED MNIST, the train dataset is equally split across two clients and the entire test set is attributed to the test client. Preliminary labels for binary classification are constructed such $\tilde{y} = 0$ for circle and $\tilde{y} = 1$ for square. Next, we add noise to the preliminary label by flipping $\tilde{y}$ with a probability $\delta_k = 0.25, \forall k$ to construct the final label $y$. We next flip the final label with a probability $p_k$ to designate a color ($z$), with $p_1 = 0.2$ for the first client, $p_2 = 0.1$ for the second

---
[2]https://github.com/deepmind/dsprites-dataset

client and $p_3 = 0.9$ for the test client. The image is colored red, if $z = 1$ and green if $z = 0$. A sample batch of images elucidating the above construction is shown in Figure 6.

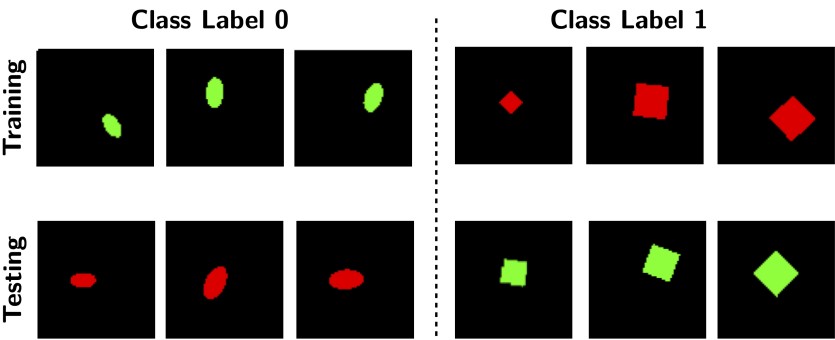

Figure 6: SPURIOUS CIFAR10: Illustration of samples containing high spurious correlation between labels and color during training. While testing, this correlation is significantly reduced as label 0 is now highly correlated with the color corresponding to label 1 (while training) and vice-versa.

### C.5   Extended Datasets

We extend the two datasets COLORED MNIST, and COLORED FASHION MNIST as in Choe et al. [2020] to robustly test our approach across multiple clients. Analogous to COLORED MNIST with two training environments ($N = 2$), we extend the datasets to incorporate $N = 2, 3, 5$ and $10$ training clients. In particular, we attribute each client with a unique flipping probability $p_k$. For each value of $N$, the maximum value of $p_k, \forall k$ is 0.3, while the minimum value is 0.1. The values of $p_k$ for each client are spaced evenly between this range. For example, for the case of $N = 5$ clients, the flipping probabilities of clients are $p_1 = 0.3$, $p_2 = 0.2$, and $p_3 = 0.1$. The flipping probability $\delta_k$ which decides the final label $y$ is fixed to 0.25 for all clients. The maximum and the minimum values of $p_k$ are chosen in a way that the average spurious correlation which is 0.8 is more than the invariant correlation i.e. 0.75.

Further, since all the previous settings were binary classification tasks, we extend the standard datasets COLORED MNIST and COLORED FASHION MNIST over multi class classification Choe et al. [2020]. Specifically, we extend the number of classes from 2 to 5 and 10. Specifically, a unique color is assigned to each output class such that the label is highly correlated ($\sim 80\% - 90\%$) with the color in the training set. In the test set, these correlations are significantly reduced ($10\%$) by allowing high spurious correlations with the color of the following class. For instance, in testing, the color corresponding to class label 8 would be the one which was heavily corrected with class label 9 while training. This reduces the original spurious correlations and is hence useful for evaluating the extent to which the trained model has learned the invariant features. A sample batch of images elucidating the above construction is shown in Figure 7.

We do not construct a multi-class classification setup for SPURIOUS CIFAR10 and COLORED DSPRITES. For the former, it is difficult to find a unique spatial location in the image corresponding to each class (e.g. in case of 10-class classification). For the latter, the Desprites dataset can be classified into three categories on the basis of shape present in the image: square, ellipse, and heart. Hence, introducing more than three classes requires categorization of other latent factors like orientation, scale or position.

## D   Experimental Setup

### D.1   Architecture Details

For all the approaches using a fixed representation i.e. F-FL GAMES, F-FL GAMES (SMOOTH), *parallelized* F-FL GAMES and *parallelized* F-FL GAMES (SMOOTH), we use the architecture mentioned below. The architecture used to train a predictor at each client is a multi-layered perceptron with three fully connected (FC) layers. The details of layers are as follows:

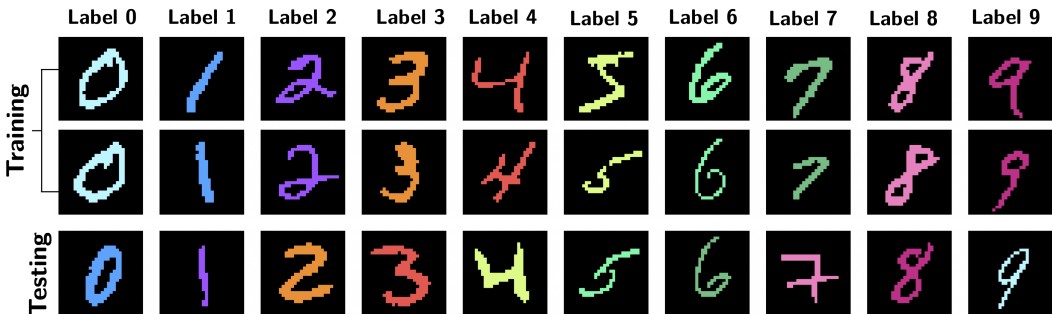

Figure 7: EXTENDED COLORED MNIST: Illustration of samples corresponding to a 10-digit classification task. During training, each class label is spuriously correlated with a unique color. While testing, this correlation is significantly reduced as each image is colored with the color corresponding to its succeeding label.

- Flatten: A flatten layer that converts the input of shape (`Batch Size, Length, Width, Depth`) into a tensor of shape (`Batch Size, Length * Width * Depth`)
- FC1: A fully connected layer with an output dimension of 390, followed by ELU non-linear activation function
- FC2: A fully connected layer with an output dimension of 390, followed by ELU non-linear activation function
- FC3: A fully connected layer with an output dimension of 2 (Classification Layer)

This same architecture is used across approaches with fixed representation.

For the approaches with trainable representation i.e. V-FL GAMES, V-FL GAMES (SMOOTH), *parallelized* V-FL GAMES and *parallelized* V-FL GAMES (SMOOTH), we use the following architecture for the representation learner

- Flatten: A flatten layer that converts the input of shape (`Batch Size, Length, Width, Depth`) into a tensor of shape (`Batch Size, Length * Width * Depth`)
- FC1: A fully connected layer with an output dimension of 390, followed by ELU non-linear activation function

The output from FC1 is fed into the following architecture, which is used as the base network to train the predictor at each client:

- FC1: A fully connected layer with an output dimension of 390, followed by ELU non-linear activation function
- FC2: A fully connected layer with an output dimension of 390, followed by ELU non-linear activation function
- FC3: A fully connected layer with an output dimension of 2 (Classification Layer)

This architecture is used across approaches with variable representation.

### D.2   Optimizer and other hyperparameters

We use a different set of hyper-parameters based on the dataset. In particular,

- COLORED MNIST and COLORED FASHION MNIST: For fixed representation, we use Adams optimizer with a learning rate of 2.5e-4 across all experiments (sequential or parallel). For the variable representation, we use Adams optimizer with a learning rate of 2.5e-5 for the representation learner and the same optimizer with a learning rate of 2.5e-4 for the predictor at each client.
- SPURIOUS CIFAR10: For fixed representation, we use Adams optimizer with a learning rate of 1.0e-4 across all experiments (sequential or parallel). For the variable representation,

we use Adams optimizer with a learning rate of 9.0e-4 for the representation learner and the same optimizer with a learning rate of 1.0e-4 for the predictor at each client.

- COLORED DSPRITES: For fixed representation, we use Adams optimizer with a learning rate of 9.5e-4 across all experiments (sequential or parallel). For the variable representation, we use Adams optimizer with a learning rate of 2.5e-4 for the representation learner and the same optimizer with a learning rate of 9.0e-4 for the predictor at each client

For all the experiments, we fix the batch size to 256 and optimize the Cross Entropy Loss. We use the same termination criterion as in Ahuja et al. [2020]. Specifically, we stop training when the observed oscillations become stable and the ensemble model is in a lower training accuracy state. We choose a training threshold and terminate the training as soon as the training accuracy drops below this value. In order to ensure stability of oscillations, we set a period of warm start. In this period, the training is not stopped even if the accuracy drops below the threshold. For variable representation, the duration of this warm start period is set to the number of training steps in an epoch i.e. (training data size/ batch size). However, for the approaches with fixed representation, this period is fixed to $N$ rounds where $N$ is the number of training clients. In particular, for two clients, the warm start period ends as soon as the second client finishes playing its optimal strategy for the first time.

### D.3 Computing Environment

The experiments were done on an Intel Xeon E5 Processor with 2133MHz DDR4 and NVIDIA Tesla v100, 32GB GPU.

## E Algorithm

Algorithm 1 represents the pseudo code for V-FL GAMES (SMOOTH+FAST). In the following analysis, the terms 'Sequential' and 'Parallel' denote BRD with clockwise playing sequences and simultaneous updates respectively (Lines 20 and 26 of Algorithm 1). We use FL GAMES as an umbrella term that constitutes all the discussed algorithmic modifications. F-FL GAMES and V-FL GAMES refer to the privacy preserving variants of FL GAMES. The approach used to smoothen out the oscillations (Line 4 of Algorithm 1) is denoted by F-FL GAMES (SMOOTH) or V-FL GAMES (SMOOTH) depending on the constraint on $\phi$. The fast variant with high convergence speed is typified as V-FL GAMES (SMOOTH+FAST)) (Line 9 of Algorithm 1).

## F Additional Results and Analysis

### F.1 Robustness to the number of Outcomes

Table 2: COLORED MNIST: Comparison of F-FL GAMES and F-FL GAMES (Parallel) with increasing number of output classes, in terms of the training and testing accuracy (mean ± std deviation). Here 'Seq.' is an abbreviation used for 'Sequential', which denotes F-FL GAMES.

| Type | # Classes | TRAIN ACCURACY | TEST ACCURACY |
|---|---|---|---|
| Seq. | 2 | 75.13 ± 1.38 | 68.40 ± 1.83 |
| | 5 | 79.39 ± 0.91 | 69.90 ± 3.16 |
| | 10 | 82.27 ± 0.76 | 69.22 ± 3.11 |
| Parallel | 2 | 71.71 ± 8.23 | 69.73 ± 2.12 |
| | 5 | 78.61 ± 2.86 | 68.42 ± 2.54 |
| | 10 | 82.17 ± 1.21 | 69.29 ± 3.17 |

As described in Section C.5, we test the robustness of our approach to an increase in number of output classes. We compare F-FL GAMES and *parallelized* F-FL GAMES across 2-digit, 5-digit and 10-digit classification for COLORED MNIST and COLORED FASHION MNIST.

As shown in Tables 2 and 3, both the sequential and the parallel version of FL GAMES i.e. F-FL GAMES and *parallelized* F-FL GAMES respectively are robust to an increase in the number of output classes. For both the datasets, *parallelized* F-FL GAMES performs at par or better than F-FL GAMES.

**Algorithm 1** *Parallelized* FL GAMES (SMOOTH+FAST)

---

1: **Notations:** $\mathcal{S}$ is the set of $N$ clients; $\mathcal{B}_k$ and $\mathcal{P}_k$ denote the buffer and information set containing copies of $\mathcal{B}_i, \forall i \neq k \in \mathcal{S}$ at client, $k$ respectively.

2: **PredictorUpdate(k):**

3:     /* Two-way ensemble game to update predictor at each client $k$ */

4:     $w_k \leftarrow \text{SGD}\Big[\ell_k\Big\{ \frac{1}{|\mathcal{E}_{tr}|}(w_k' + \sum_{\substack{q \in \mathcal{E}_{tr} \\ q \neq k}} w^q + \sum_{\substack{p \in \mathcal{E}_{tr} \\ p \neq k}} \frac{1}{|\mathcal{B}_p|} \sum_{j=1}^{|\mathcal{B}_p|} w_j^p) \circ \phi \Big\}\Big]$

5:     Insert $w_k$ to $\mathcal{B}_k$, discard oldest model in $\mathcal{B}_k$ if full

6:     return $w_k$

7: **RepresentationUpdate(k):**

8:     /* Gradient Descent (GD) over entire local dataset at client $k$ */

9:     **for** every batch $b \in \mathcal{B}$ **do**

10:         Compute $\nabla \ell_k(w_{\text{cur}}^{\text{av}} \circ \phi_{\text{cur}}; b)$; Add in $\nabla \phi_k$

11:     return $\nabla \phi_k$

12: **Server executes:**

13:     Initialize $w_k, \forall k \in \mathcal{S}$ and $\phi$

14:     **while** round $\leq$ max-round **do**

15:         /* Update representation $\phi$ at even round parity */

16:         **if** round is even **then**

17:             **if** Fixed-Phi **then**

18:                 $\phi_{\text{cur}} = I$

19:             **if** Variable-Phi **then**

20:                 **for** each client $k \in \mathcal{S}$ in parallel **do**

21:                     $\nabla \phi_k = \text{RepresentationUpdate(k)}$

22:                 /* Update representation $\phi$ */

23:                 $\phi_{\text{next}} = \phi_{\text{cur}} - \eta\Big( \sum_{k \in \mathcal{S}} \frac{N_k}{\sum_{j \in \mathcal{S}} N_j} \nabla \phi_k \Big)$

24:                 $\phi_{\text{cur}} = \phi_{\text{next}}$

25:         **else**

26:             **for** each client $k \in \mathcal{S}$ in parallel **do**

27:                 $w_k \leftarrow \text{PredictorUpdate(k)}$

28:             /* Client $k$ updates its information set $\mathcal{P}_k$ by updating copies of predictors of other clients */

29:             Communicate $\forall k, \mathcal{P}_k \leftarrow \{w_i, \forall i \neq k \in S\}$

30:         round $\leftarrow$ round + 1

31:     $w_{\text{curr}}^{\text{av}} = \frac{1}{N} \sum_{k \in \mathcal{S}} w_{\text{curr}}^k$

---

Table 3: COLORED FASHION MNIST: Comparison of F-FL GAMES and F-FL GAMES (Parallel) with increasing number of output classes, in terms of the training and testing accuracy (mean $\pm$ std deviation). Here 'Seq.' is an abbreviation used for 'Sequential', which denotes F-FL GAMES.

| Type | # Classes | TRAIN ACCURACY | TEST ACCURACY |
|------|-----------|----------------|---------------|
| Seq. | 2 | $50.36 \pm 2.78$ | $47.36 \pm 4.33$ |
| | 5 | $77.28 \pm 1.54$ | $69.35 \pm 0.66$ |
| | 10 | $80.02 \pm 0.38$ | $71.22 \pm 3.00$ |
| Parallel | 2 | $55.06 \pm 2.04$ | $52.07 \pm 1.60$ |
| | 5 | $77.61 \pm 1.51$ | $70.83 \pm 0.96$ |
| | 10 | $80.12 \pm 0.78$ | $70.39 \pm 2.28$ |

## F.2 Effect of Simultaneous BRD

Similar to the experiments conducted for COLORED MNIST, where we compared F-FL GAMES and *parallelized* F-FL GAMES across an increase in the number of clients, we replicate the same setup for COLORED FASHION MNIST, SPURIOUS CIFAR10 and COLORED DSPRITES. We report the results on the three datasets in Figures 8, 9 and 10. Consistent with the results on COLORED MNIST, as the number of clients in the FL system increases, there is a sharp increase in the number of communication rounds required to reach equilibrium. However, the same doesn't hold true for *parallelized* F-FL GAMES. Further, the accuracy achieved by *parallelized* F-FL GAMES is comparable or better than that achieved by F-FL GAMES. Moreover, as observed from Table 4,

Table 4: COLORED FASHION MNIST, SPURIOUS CIFAR10 and COLORED DSPRITES: Comparison of convergence of methods in terms of mean number of rounds required to reach equilibrium.

| | | ALGORITHM | Number of Communication Rounds | | |
| --- | --- | --- | --- | --- | --- |
| | | | COLORED FASHION MNIST | SPURIOUS CIFAR10 | COLORED DSPRITES |
| Fixed | Seq. | F-FL GAMES | 158.0 | 339.5 | 367.9 |
| | | F-FL GAMES (SMOOTH) | 113.2 (1.4×) | 304.0 (1.1×) | 257.0 (1.4×) |
| | Par. | F-FL GAMES | 134.2 (1.2×) | 237.4 (1.4×) | 233.1 (1.6×) |
| | | F-FL GAMES (SMOOTH) | **58.4** (2.7×) | **217.6** (1.6×) | **210.4** (1.8×) |
| Variable | Seq. | V-FL GAMES | 454.8 | 557.6 | 628.6 |
| | | V-FL GAMES (SMOOTH+FAST) | 189.0 (2.4×) | 363.3 (1.5×) | 455.5 (1.4×) |
| | Par. | V-FL GAMES | 367.3 (1.2×) | 498.7 (1.1×) | 520.9 (1.2×) |
| | | V-FL GAMES (SMOOTH+FAST) | **127.8** (3.6×) | **102.4** (5.4×) | **203.0** (3.1×) |

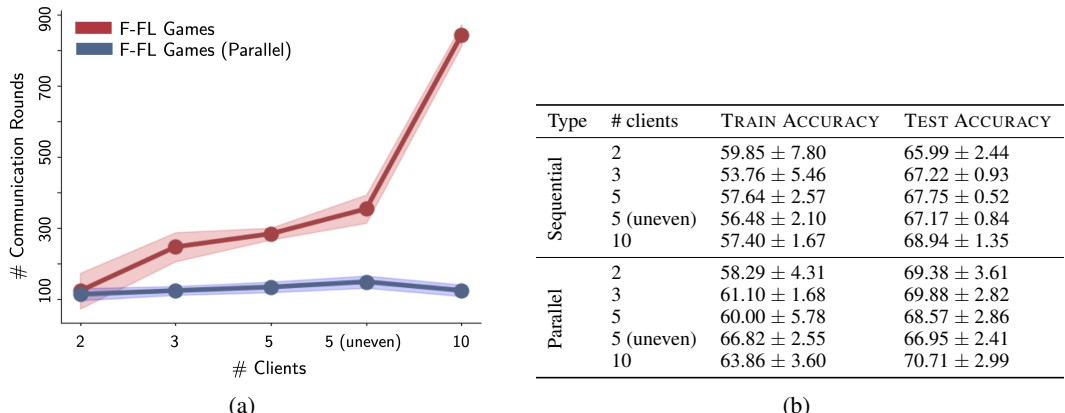

| Type | # clients | TRAIN ACCURACY | TEST ACCURACY |
| --- | --- | --- | --- |
| Sequential | 2 | $59.85 \pm 7.80$ | $65.99 \pm 2.44$ |
| | 3 | $53.76 \pm 5.46$ | $67.22 \pm 0.93$ |
| | 5 | $57.64 \pm 2.57$ | $67.75 \pm 0.52$ |
| | 5 (uneven) | $56.48 \pm 2.10$ | $67.17 \pm 0.84$ |
| | 10 | $57.40 \pm 1.67$ | $68.94 \pm 1.35$ |
| Parallel | 2 | $58.29 \pm 4.31$ | $69.38 \pm 3.61$ |
| | 3 | $61.10 \pm 1.68$ | $69.88 \pm 2.82$ |
| | 5 | $60.00 \pm 5.78$ | $68.57 \pm 2.86$ |
| | 5 (uneven) | $66.82 \pm 2.55$ | $66.95 \pm 2.41$ |
| | 10 | $63.86 \pm 3.60$ | $70.71 \pm 2.99$ |

(a)                                                          (b)

Figure 8: COLORED FASHION MNIST: (a) Number of communication rounds required to achieve the Nash equilibrium versus the number of clients in the FL setup; (b) Comparison of F-FL GAMES and F-FL GAMES (Parallel) with an increase in the number of clients, in terms of training and testing accuracy (mean $\pm$ std deviation).

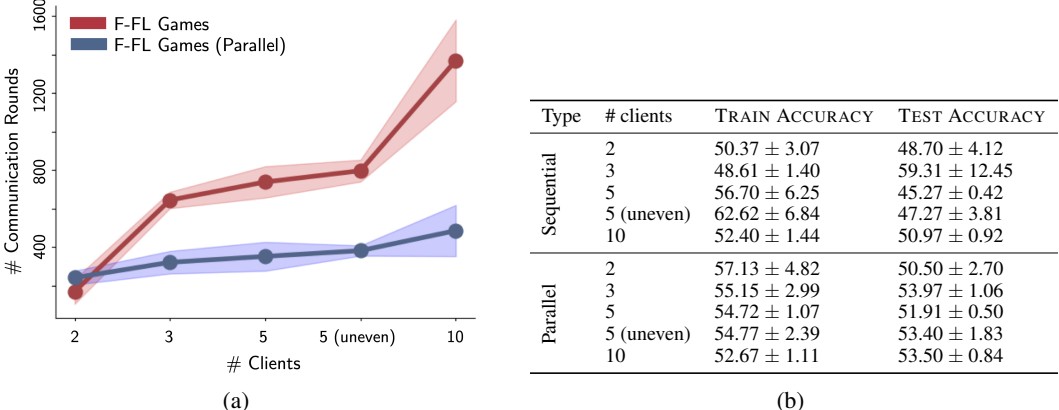

| Type | # clients | TRAIN ACCURACY | TEST ACCURACY |
| --- | --- | --- | --- |
| Sequential | 2 | $50.37 \pm 3.07$ | $48.70 \pm 4.12$ |
| | 3 | $48.61 \pm 1.40$ | $59.31 \pm 12.45$ |
| | 5 | $56.70 \pm 6.25$ | $45.27 \pm 0.42$ |
| | 5 (uneven) | $62.62 \pm 6.84$ | $47.27 \pm 3.81$ |
| | 10 | $52.40 \pm 1.44$ | $50.97 \pm 0.92$ |
| Parallel | 2 | $57.13 \pm 4.82$ | $50.50 \pm 2.70$ |
| | 3 | $55.15 \pm 2.99$ | $53.97 \pm 1.06$ |
| | 5 | $54.72 \pm 1.07$ | $51.91 \pm 0.50$ |
| | 5 (uneven) | $54.77 \pm 2.39$ | $53.40 \pm 1.83$ |
| | 10 | $52.67 \pm 1.11$ | $53.50 \pm 0.84$ |

(a)                                                          (b)

Figure 9: SPURIOUS CIFAR10: (a) Number of communication rounds required to achieve the Nash equilibrium versus the number of clients in the FL setup; (b) Comparison of F-FL GAMES and F-FL GAMES (Parallel) with an increase in the number of clients, in terms of training and testing accuracy (mean $\pm$ std deviation).

across the three benchmarks, all variants of parallel FL GAMES show significantly faster convergence compared to their corresponding sequential counterparts. This improvement is as large as 3.6× when comparing *parallelized* V-FL GAMES (SMOOTH+FAST) and *sequential* V-FL GAMES (SMOOTH+FAST).

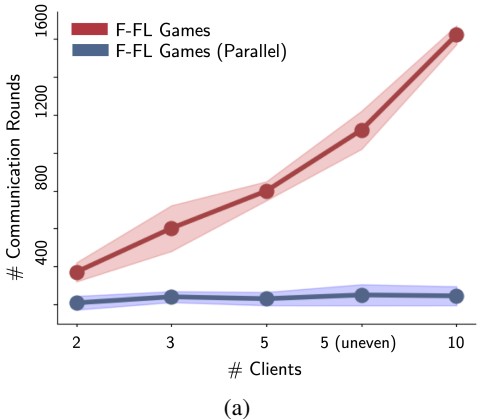

| Type | # clients | TRAIN ACCURACY | TEST ACCURACY |
|---|---|---|---|
| Sequential | 2 | 51.12 ± 4.15 | 54.10 ± 3.45 |
| | 3 | 53.21 ± 3.88 | 55.92 ± 6.22 |
| | 5 | 52.70 ± 4.67 | 54.60 ± 5.34 |
| | 5 (uneven) | 54.84 ± 2.63 | 55.19 ± 3.03 |
| | 10 | 53.75 ± 4.16 | 54.89 ± 4.20 |
| Parallel | 2 | 52.31 ± 2.78 | 57.30 ± 4.02 |
| | 3 | 53.11 ± 3.69 | 56.04 ± 5.20 |
| | 5 | 53.06 ± 2.94 | 57.01 ± 4.23 |
| | 5 (uneven) | 52.30 ± 4.05 | 58.05 ± 5.73 |
| | 10 | 53.14 ± 3.97 | 59.35 ± 5.07 |

|     (a)     |     (b)     |
|---|---|

Figure 10: COLORED DSPRITES: (a) Number of communication rounds required to achieve the Nash equilibrium versus the number of clients in the FL setup; (b) Comparison of F-FL GAMES and F-FL GAMES (Parallel) with an increase in the number of clients, in terms of training and testing accuracy (mean ± std deviation).

## F.3 Effect of Memory Ensemble

Similar to the experiments conducted for COLORED MNIST, where we compared F-FL GAMES and F-FL GAMES (SMOOTH), we replicate the same setup for COLORED FASHION MNIST, SPURIOUS CIFAR10 and COLORED DSPRITES. We report the results on all the three datasets in Figures 11, 12 and 13.

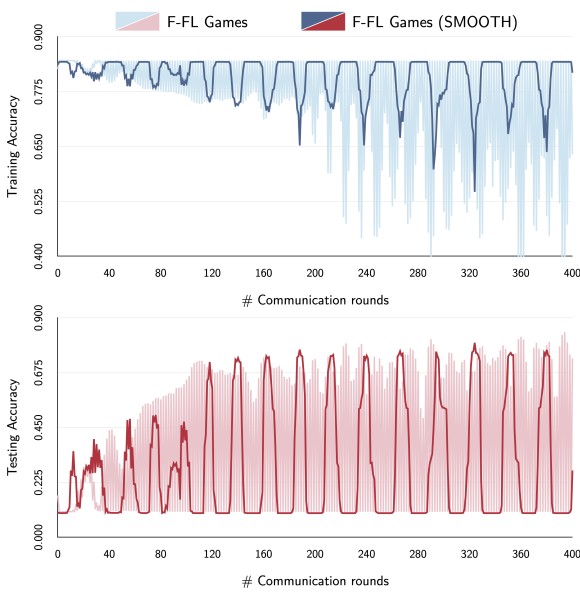

Figure 11: COLORED FASHION MNIST: Evolution of Training and Testing Training for F-FL GAMES and F-FL GAMES (SMOOTH) using a buffer size of 5, over the number of communication rounds

Consistent with the results on COLORED MNIST, performance curves oscillate at each step for F-FL GAMES while the oscillations in F-FL GAMES (SMOOTH) are observed after an interval of roughly 40 rounds for all the three datasets. Further, F-FL GAMES (SMOOTH) also achieves high testing accuracy. This implies that it does not rely on the spurious features to make predictions. As observed from Table 4, apart from achieving high accuracy, both F-FL GAMES (SMOOTH) and *parallelized* F-FL GAMES (SMOOTH) require significantly fewer communication rounds as compared to F-FL GAMES and *parallelized* F-FL GAMES respectively. Moreover, *parallelized* F-FL GAMES (SMOOTH) exhibits fastest convergence rate across all the variants of fixed FL GAMES. Similar

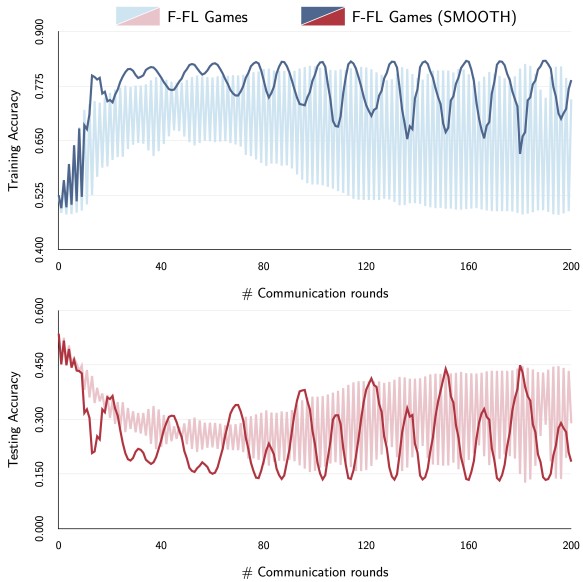

Figure 12: SPURIOUS CIFAR10: Evolution of Training and Testing Training for F-FL GAMES and F-FL GAMES (SMOOTH) using a buffer size of 5, over the number of communication rounds

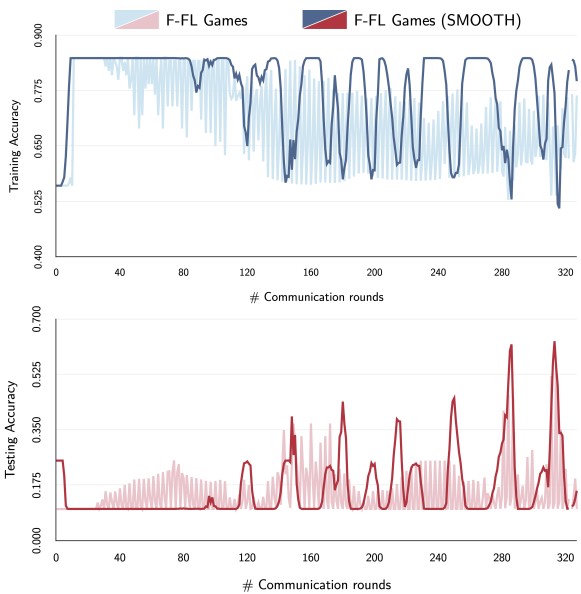

Figure 13: COLORED DSPRITES: Evolution of Training and Testing Training for F-FL GAMES and F-FL GAMES (SMOOTH) using a buffer size of 5, over the number of communication rounds

performance evolution curves are also observed for *parallelized* F-FL GAMES (SMOOTH) with an added benefit of faster convergence as compared to F-FL GAMES and F-FL GAMES (SMOOTH).

### F.4 Effect of using Gradient Descent (GD) for $\phi$

Similar to the experiments conducted for COLORED MNIST, where we compared V-FL GAMES and V-FL GAMES (SMOOTH+FAST), we replicate the same setup for COLORED FASHION MNIST, SPURIOUS CIFAR10 and COLORED DSPRITES. We report the results on all the three datasets in Figures 14, 15 and 16. Consistent with the results on COLORED MNIST, V-FL GAMES (SMOOTH +FAST) is able to achieve significantly higher testing accuracy in fewer communication rounds as compared to V-FL GAMES on all the three benchmarks. Further, as evident from Table 4, *parallelized*

V-FL GAMES (SMOOTH+FAST) has the fastest convergence rate across both SPURIOUS CIFAR10 and COLORED DSPRITES. On a rather simpler dataset i.e. COLORED FASHION MNIST, *parallelized* V-FL GAMES (SMOOTH+FAST) exhibits fastest performance across all the variable variants of FL GAMES.

As discussed before, the performance of variable FL GAMES is superior to that of it's fixed variant over complex and large-scale datasets like COLORED DSPRITES. Additionally, as conspicuous from Table 4, *parallelized* V-FL GAMES (SMOOTH+FAST) requires significantly fewer communication rounds to reach the Nash equilibrium. This further underscores the importance of *parallelized* V-FL GAMES both in terms of (i) it's ability to recover the causal mechanisms of the targets, while also providing robustness to distribution shift across clients; (ii) communication efficiency.

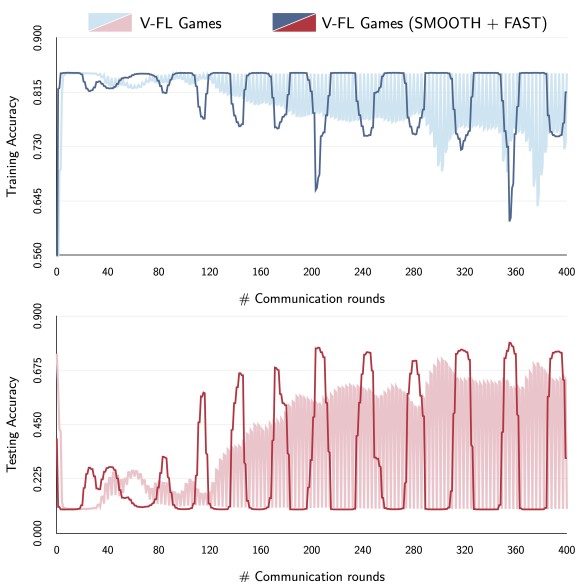

Figure 14: COLORED FASHION MNIST: Evolution of Training and Testing Training for V-FL GAMES and V-FL GAMES (SMOOTH+FAST) using a buffer size of 5, over the number of communication rounds

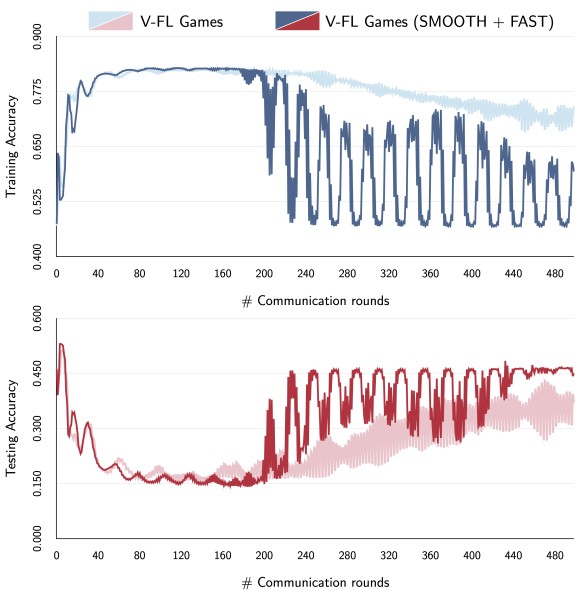

Figure 15: SPURIOUS CIFAR10: Evolution of Training and Testing Training for V-FL GAMES and V-FL GAMES (SMOOTH+FAST) using a buffer size of 5, over the number of communication rounds

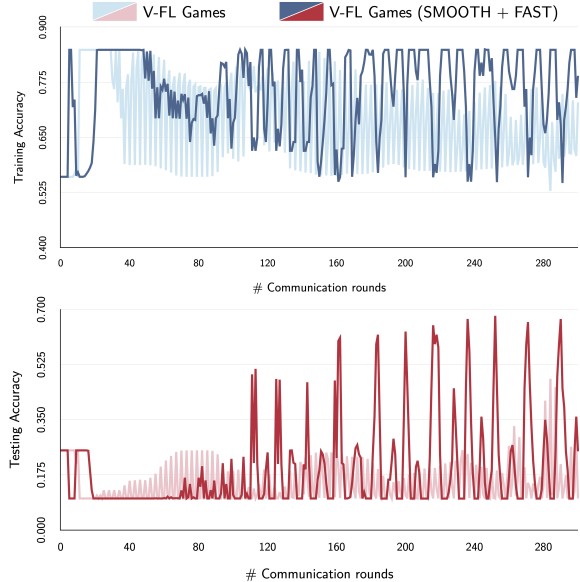

Figure 16: COLORED DSPRITES: Evolution of Training and Testing Training for V-FL GAMES and V-FL GAMES (SMOOTH+FAST) using a buffer size of 5, over the number of communication rounds

## F.5 Effect of exact best response

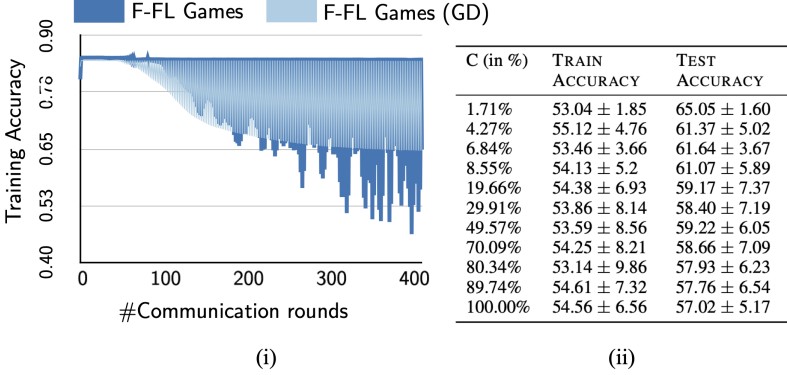

| C (in %) | TRAIN ACCURACY | TEST ACCURACY |
|---|---|---|
| 1.71% | $53.04 \pm 1.85$ | $65.05 \pm 1.60$ |
| 4.27% | $55.12 \pm 4.76$ | $61.37 \pm 5.02$ |
| 6.84% | $53.46 \pm 3.66$ | $61.64 \pm 3.67$ |
| 8.55% | $54.13 \pm 5.2$ | $61.07 \pm 5.89$ |
| 19.66% | $54.38 \pm 6.93$ | $59.17 \pm 7.37$ |
| 29.91% | $53.86 \pm 8.14$ | $58.40 \pm 7.19$ |
| 49.57% | $53.59 \pm 8.56$ | $59.22 \pm 6.05$ |
| 70.09% | $54.25 \pm 8.21$ | $58.66 \pm 7.09$ |
| 80.34% | $53.14 \pm 9.86$ | $57.93 \pm 6.23$ |
| 89.74% | $54.61 \pm 7.32$ | $57.76 \pm 6.54$ |
| 100.00% | $54.56 \pm 6.56$ | $57.02 \pm 5.17$ |

(i)                                   (ii)

Figure 17: COLORED MNIST: (i) Effect on Training accuracy of doing a gradient descent on each client for updating the predictor versus the standard training paradigm i.e. F-FL GAMES;(ii) Impact of increasing the number of local steps (C) for updating the predictor on the training and testing accuracy (mean ± std deviation).

FEDAVG provides the flexibility to train communication efficient and high quality models by allowing more local computation at each client. This is particularly detrimental in scenarios with poor network connectivity, wherein communicating at every short time span is infeasible. Inspired by FEDAVG, we study the effect of increasing the amount of local computation at each client. Specifically, in F-FL GAMES, each client updates its predictor based on a step of stochastic gradient descent over its mini-batch. We modify this setup by allowing each client to run a few steps of stochastic gradient descent locally ($C\%$). When the number of local steps at each client reaches is maximum (training data size/ mini-batch size) or $C = 100\%$), the scenario becomes equivalent to a gradient descent (GD) over the training data. For COLORED MNIST, it is evident from Figure **??**(ii) that as the number of local steps increase i.e. each client **exactly** best responds to its opponents, the testing accuracy at equilibrium starts to decrease. When the local computation reaches 100%, i.e. each client updates its local predictor based on a GD over its data, F-FL GAMES exhibits converges (as shown in Figure 17(i)).

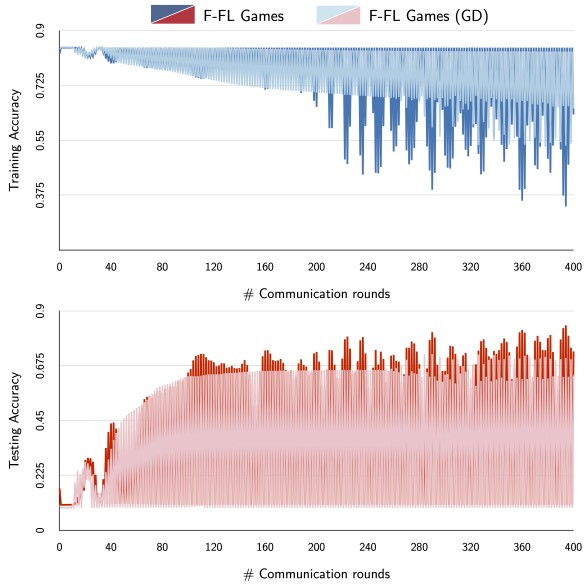

Figure 18: COLORED FASHION MNIST: Evolution of Training and Testing Training when doing a gradient descent to update the predictor at each client versus the standard training paradigm i.e. F-FL GAMES, over the number of communication rounds.

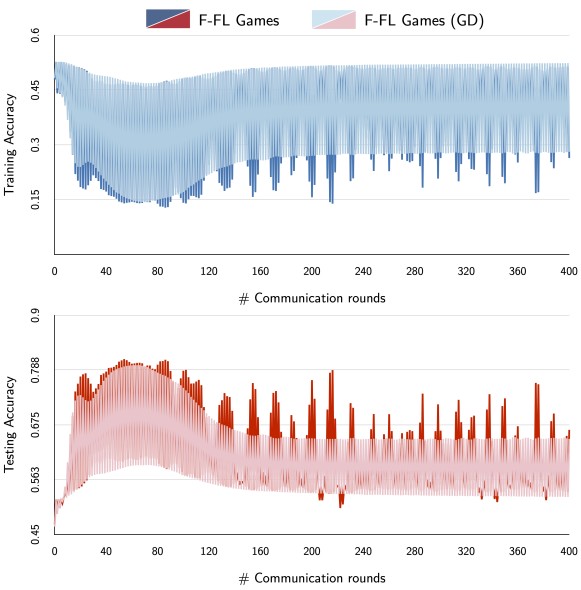

Figure 19: SPURIOUS CIFAR10: Evolution of Training and Testing Training when doing a gradient descent to update the predictor at each client versus the standard training paradigm i.e. F-FL GAMES, over the number of communication rounds.

Similar to the experiments conducted for COLORED MNIST, where we studied the influence of increasing the amount of local computation at each client, we replicate the setup for COLORED FASHION MNIST, SPURIOUS CIFAR10 and COLORED DSPRITES. Consistent with the results on COLORED MNIST and as shown in Figures 18, 19 and 20, even when the number of local steps at each client reaches its maximum i.e. (training data size/ mini-batch size)), the trained models are able to achieve high testing accuracy at equilibrium. Further, this setup achieves faster convergence with higher training accuracy at equilibrium. As observed from the empirical results and discussed by Ahuja et al. [2021], FL GAMES is guaranteed to exhibit convergence and good out-of-distribution generalization behavior Ahuja et al. [2021] despite increasing local computations. Although the

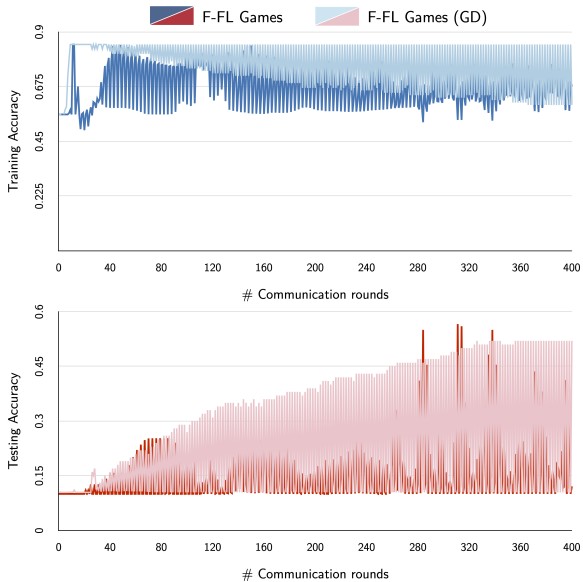

Figure 20: COLORED DSPRITES: Evolution of Training and Testing Training when doing a gradient descent to update the predictor at each client versus the standard training paradigm i.e. F-FL GAMES, over the number of communication rounds.

testing accuracy at convergence is lower compared to the standard setup, this approach opens avenues for practical deployment of the approach in FL.

