# OpenReview forum: "FL Games: A Federated Learning Framework for Distribution Shifts"
_NeurIPS.cc/2022/Workshop/Federated_Learning — FL-NeurIPS 2022 Oral_

### Official Review · Reviewer_uesy · 2022-10-10
**invariant risk minimization for learning better causal representations in FL**

In order for models trained in a FL fashion to generalize well, it is important that the models are not fooled by spurious correlations in the training data of individual clients. This work claims that methods designed to address data-heterogeneity (such as Scaffold, FedNova, etc.) fail on such tasks and presents a new method (F-FL) that addresses this issue.

The presented method is derived from recent advances on solving Invariant Risk Minimization (IRM) problems and can be seen as a proposed distributed/FL implementation of IRM GAMES.

Numerical evidence is presented on synthetic benchmark datasets. The experiments show that F-FL significantly outperforms standard methods such as FedAvg, FedProx, etc.), although a comparison to the aforementioned Scaffold, FedNova, etc. typo of methods is missing.

I think this is a very good contribution to the workshop. While certain aspects in experiment design (missing baselines, synthetic data) could cause some discussion in the community, the workshop might be an ideal platform for this.

---

### Official Review · Reviewer_xNsc · 2022-10-16
**Offers a new game theoretic perspective, however no significantly new insights are provided and the experimental setup is limited.**

## Summary:
The paper discusses about a game theoretic formulation for FL with the goal of learning invariant feature representations. The authors base the work on Invariant Risk Minimization Games (IRM) which aims at learning invariant representations across a set of tasks (or data distributions). The authors point out some shortcomings of IRM games (in general and in the FL setup) and propose some minor tweaks to address those challenges. They evaluate the algorithm on some toy datasets and provide comparison against simple baselines such as FedAvg.

The writing is clear despite some minor issues (described at the end). The game theoretic perspective is new and shows the issue of oscillations while using the best response dynamics however, the work really doesn't address the issue completely. There is no theoretical analysis of the algorithm. The novelty of the work is very limited as the algorithm and the framework is a simple extension of the IRM games work. The only thing different is the update of $\phi$ and using the best response update for $w_k$ from the past history. The proposed method also doesn't solve the problems with IRM games completely. The evaluation of the algorithm is also limited and a better evaluation on more realistic setting can improve the quality of the work.

## Questions:
 - Can it be possible that the same $\phi$ is not optimal for all environments? For example, since different environments can have different sensors, the input space can also vary across each environment/task. Therefore, same $\phi$ can not work. More discussion on this aspect would be useful, such as how hardware heterogeneity can be handled.
- What space does $g_k$ lie in? In Section 3.1, $g_k$ is seen as the gradient of the risk. Is it with respect to all the model parameters ($w, \phi$) or only the feature extractor $\phi$? In the update equation of $\phi_t$, $g_k$ is used as the gradient. So, is $g_k$ computed by keeping $\{w_k\}$ fixed for all workers?
- How does this work relate to [3]? They also train the final layer weights $w_k$ by keeping the global representations fixed, and then update the representations separately.
- Also, generally explaining where the algorithm differs from simple FedAvg could be a useful way to illustrate the algorithm. As far as I understand, it differs form FedAvg in keeping the updates for $\phi$ and $\{w_k\}$ separate where each local $\{w_k\}$ is trained for a fixed $\phi$ that is updated separately. Please correct me if my understanding is wrong.

## Concerns:
- The paper introduces the game theoretic framework and the algorithm, but they do not show any theoretical analysis of the method such as convergence or generalization properties. The authors mention in the contributions that the convergence rate of IRM is slow and they improve it using their method, however no mention of the actual convergence properties are discussed. As the results show oscillations, I doubt that the convergence is hard to show under general non-convex scenarios. It would be interesting to throw some light on the theoretical properties on what might be the reasons for such behavior.
- This is usually fine if the paper's main focus is experimentation but here the experimental setup is severely limited for the following reasons:
  - The results are presented for the case where there are 2 clients. For example, in colored MNIST, the training dataset is split into 2 clients and the test set is used for evaluation. In real FL scenarios, there maybe 10-100 (in cross silo FL) and often 1000+ (cross device FL). Although the authors present results for multiple clients in Fig 1(b), comparison against other baselines is  not provided.
   - The baselines considered are not really known for good generalization. The authors should consider algorithms like FedGen [2] that show good generalization in non-iid FL.
   - The simulation of statistical heterogeneity is very limited. The setup used is to devise a colored MNIST where there are confounders /spurious correlations (such as the color) that are not invariant with respect to the digit classification task. While this is a good setup to demonstrate invariant risk minimization with datasets that have spurious correlations, this is still a toy setup. How does the algorithm work against other (more practical) non-iid scenarios such as FEMNIST, CelebA datasets and $\alpha$ Dirichlet split [1]?
   - The training accuracy in Fig 2 (b, c) has higher amplitude for the smooth method as compared to the non smooth method (in fact significant drops in accuracy). But the authors do not discuss this.
   - The authors claim that smooth+fast algorithm achieves higher accuracy much faster than the other algorithm in Fig (2c) however, the figure really doesn't lead to that conclusion.
    -
 - The communication complexity of the clients is not discussed. Usually in FedAvg, the server sends just an aggregate, but here, each client needs to know the model of every other client. This leads to a very high communication complexity that scales with the number of workers, which is not desired.


## Minor:
- IRM is mentioned multiple times before actually defining what IRM is.
- The definition of the risk $R^e$ (in Def 1 and in Sections 2, 3) is not defined
- $w_{av}$ should not be in the outer minimization in (1) since $w_{av}$ is defined as the average of local models
- The notation and terminology is inconsistent. For example, the client index is used as subscript and superscript in different locations ($w_k^{'}$ is used in the optimization objective, whereas $w^k$ is used for client models). What is V-FL or F-FL, what do those letters stand for (some background may be helpful)?



[1] Hsu, T.-M. H., Qi, H., and Brown, M. (2019). Measuring the effects of non-identical data distribution
for federated visual classification. arXiv preprint arXiv:1909.06335

[2] Zhu, Z., Hong, J., and Zhou, J. (2021). Data-free knowledge distillation for heterogeneous federated learning. arXiv preprint arXiv:2105.10056

[3] Pillutla et al. (ICML 2022) Federated Learning with Partial Model Personalization

---

### Official Review · Reviewer_JH6o · 2022-10-18
**accept**

This work proposed a new framework (FL games) that learns invariant causal features across client in a federated learning setting that have good out-of-distribution generalization on new train clients or test clients unseen during training. The underlying game theoretic framework was based on Ahuja et al. [2020] (IRM games). The paper examines the training dynamics and complexity for IRM games, and show that using ensembles over client’ historical actions can smooth out the oscillations in training.

Pro:
Federated learning with non-iid clients is a both practical and important problem, with many downstream applications (e.g. medical data). I believe that the proposed method in this paper would be of interest to the workshop community. The paper conducted a good set of comparative experiments which evaluates the proposed BRD algorithm with existing benchmarks.

Cons:
The analysis on why smoothing improves upon the best response dynamics is rather limited, especially its implications on the game-theoretic framework. It would be good to see a more detailed analysis on how the smoothing trick addresses the high frequency oscillations under various scenarios.

---

> ### Public Comment · (anonymous) · 2022-12-17
> **Thanks**
>
> Thanks for the review.

---

### Decision · Program_Chairs · 2022-10-20

Accept (Oral)